# Borylated 2,3,4,5-Tetrachlorophthalimide and Their 2,3,4,5-Tetrachlorobenzamide Analogues: Synthesis, Their Glycosidase Inhibition and Anticancer Properties in View to Boron Neutron Capture Therapy

**DOI:** 10.3390/molecules27113447

**Published:** 2022-05-26

**Authors:** David M. Campkin, Yuna Shimadate, Barbara Bartholomew, Paul V. Bernhardt, Robert J. Nash, Jennette A. Sakoff, Atsushi Kato, Michela I. Simone

**Affiliations:** 1Discipline of Chemistry, University of Newcastle, Callaghan, NSW 2308, Australia; campkindavid@gmail.com; 2Priority Research Centre for Drug Development, University of Newcastle, Callaghan, NSW 2308, Australia; jennette.sakoff@newcastle.edu.au; 3Department of Hospital Pharmacy, University of Toyama, 2630 Sugitani, Toyama 930-0194, Japan; m1961224@ems.u-toyama.ac.jp (Y.S.); kato@med.u-toyama.ac.jp (A.K.); 4Phytoquest Ltd., Plas Gogerddan, Aberystwyth, Ceredigion SY23 3EB, UK; barbara.bartholomew@phytoquest.co.uk (B.B.); robert.nash@phytoquest.co.uk (R.J.N.); 5School of Chemistry & Molecular Biosciences, University of Queensland, Brisbane, QLD 4072, Australia; p.bernhardt@uq.edu.au; 6Calvary Mater Newcastle Hospital, Edith Street, Waratah, NSW 2298, Australia

**Keywords:** boron, phthalimide, benzamide, glycosidase, cancer, boron neutron capture therapy

## Abstract

Tetrachlorinated phthalimide analogues bearing a boron-pinacolate ester group were synthesised via two synthetic routes and evaluated in their glycosidase modulating and anticancer properties, with a view to use them in boron neutron capture therapy (BNCT), a promising radiation type for cancer, as this therapy does little damage to biological tissue. An unexpected decarbonylation/decarboxylation to five 2,3,4,5-tetrachlorobenzamides was observed and confirmed by X-ray crystallography studies, thus, giving access to a family of borylated 2,3,4,5-tetrachlorobenzamides. Biological evaluation showed the benzamide drugs to possess good to weak potencies (74.7–870 μM) in the inhibition of glycosidases, and to have good to moderate selectivity in the inhibition of a panel of 18 glycosidases. Furthermore, in the inhibition of selected glycosidases, there is a core subset of three animal glycosidases, which is always inhibited (rat intestinal maltase α-glucosidase, bovine liver β-glucosidase and β-galactosidase). This could indicate the involvement of the boron atom in the binding. These glycosidases are targeted for the management of diabetes, viral infections (via a broad-spectrum approach) and lysosomal storage disorders. Assays against cancer cell lines revealed potency in growth inhibition for three molecules, and selectivity for one of these molecules, with the growth of the normal cell line MCF10A not being affected by this compound. One of these molecules showed both potency and selectivity; thus, it is a candidate for further study in this area. This paper provides numerous novel aspects, including expedited access to borylated 2,3,4,5-tetrachlorophthalimides and to 2,3,4,5-tetrachlorobenzamides. The latter constitutes a novel family of glycosidase modulating drugs. Furthermore, a greener synthetic access to such structures is described.

## 1. Introduction

The phthalimide scaffold appears in several drugs, including the fungicide *N*-(trichloromethylthio)phthalimide (Folpet^®^), thalidomide—now used for leprosy—and in the first line treatment of multiple myeloma [1], the antibacterial talmetoprim, the antifungal amphotalide, and the antiepileptic taltrimide (Figure 1) [2].

Phthalimide analogues have also been shown to display glycosidase inhibition [3,4,5], with the 2,3,4,5-tetrachlorophthalimide scaffold deemed necessary for potent activity and the corresponding unsubstituted phthalimide derivatives showing reduced activity [6].

In our group, we are interested in the use of organic boron as a pharmacophoric group in its boronic acid (R-B(OH)_2_) and boronate ester (R-B(OR′)_2_) [7,8] functional groups and in the development of synthetic methodologies for the installation of this pharmacophore on biologically active molecules to study and expand the palette of enzyme–drug interactions [9,10,11,12,13].

Glycosidase enzymes are involved in a number of disease states, ranging from diabetes and lysosomal storage disorders to viral infections, with their modulation being paramount in the management of these diseases [14,15,16,17,18,19,20,21,22]. The introduction of boron atoms to drug molecules also provides access to potential boron neutron capture therapy (BNCT) agents. BNCT provides the opportunity to utilise a type of radiotherapy that causes minimal damage to healthy tissue [23,24,25].

We report the synthesis of a family of novel drugs consisting of borylated 2,3,4,5-tetrachlorophthalimides and 2,3,4,5-tetrachlorobenzamides. The latter group arose from a decarbonylation side reaction, giving expedited access to them. Drugs have been characterized, including by X-ray crystallographic analysis in two instances. These confirmed the structural integrity, the outcome of the side reaction and the conformation of the boronate ester groups in the solid state. Furthermore, biological assays against glycosidase enzymes and cancer cell lines highlighted a good inhibitor for bovine liver β-galactosidase and three potent growth inhibitors and, of these, one selective growth inhibitor for cancer versus healthy cell lines in the cancer assay. These drugs represent an optimal set for further derivatisations.

## 2. Results and Discussion

### 2.1. Summary of Synthetic Work

Synthesis of the *N*-borylated 2,3,4,5-tetrachlorophthalimides was attempted via two synthetic strategies: the double acyl substitution route (reaction of **1** with **meta 2**, **para 2**, **ortho 4**, **meta 4** and **para 4**, Figure 1) and the S_N_2 route (reaction of **6** with **ortho 7** and **para 7**).

Synthesis of the 2,3,4,5-tetrachlorophthalimides **meta 3** and **para 3** was achieved in moderate to good yields in one synthetic step via double acyl substitution from 2,3,4,5-tetrachlorophthalic anhydride **1,** which was reacted with 3- and 4-(aminomethyl)phenylboronic acid pinacol ester hydrochloride **meta 2** and **para 2** in DMF at 85 °C. Reagent 2-(aminomethyl)phenylboronic acid pinacol ester hydrochloride was not commercially available for synthesis of **ortho 3** (not shown).

An unexpected decarbonylation reaction gave products **ortho 5**, **meta 5** and **para 5**, and decarboxylation reaction gave **ortho 8** and **para 8**.

### 2.2. Decarbonylation Reaction

To our knowledge, there are limited literature reports to the synthesis of 2,3,4,5-tetrachlorobenzyl scaffolds. Two main synthetic strategies were employed.

One method involves the synthesis of 2,3,4,5-tetrachlorophthalic acid diesters from direct esterification of 2,3,4,5-tetrachlorophthalic anhydride with primary alcohols at temperatures of 200 °C or above, which is accompanied by decarbonylation to give the expected tetrachlorophthalic acid diester and the decarbonylated 2,3,4,5-tetrachlorobenzoate ester in a ~2:1 ratio. [26] When the reaction is base catalysed (potassium carbonate, 3.63 mol%), the ratio of products is reversed (~1:2), with the decarbonylated product forming in greater amounts. The same authors also synthesised 2,3,4,5-tetrachlorobenzoic acid from 2,3,4,5-tetrachlorophthalic acid or anhydride by reaction in water, catalysed by sodium hydroxide (1 eq) at 200 °C for 7 h (93%). Then, synthesis of the corresponding secondary amides (2,3,4,5-tetrachlorobenzanilide and 2,3,4,5-tetrachlorobenzamide) is achieved in two further steps by derivatising 2,3,4,5-tetrachlorobenzoic acid to the corresponding acid chloride, then by reaction with aniline and ammonium hydroxide, respectively, in excellent overall yield.

A second method, described by Harvey et al., achieves the synthesis of the same set of target molecules via a two-/three-step sequence from aromatic starting materials (e.g., toluene, ethylbenzene), firstly via perchlorination with chlorine bubbling through the aromatic starting material, iron powder and anhydrous ferric chloride boiling in carbon tetrachloride for 8 h, followed by reaction of the perchlorinated aromatics with sulfuric acid at 180–200 °C for heptachlorotoluene or 260–280 °C for nonachloroethylbenzene starting material used. The reaction with the nonachloroethylbenzene required a further step, namely the reaction with potassium permanganate in 2N sodium hydroxide at 80 °C. The yields for 2,3,4,5-tetrachlorobenzoic acid were 29% and 14%, respectively [27].

Earlier synthetic strategies include the reaction of tetrachlorophthalic acid in a sealed vessel at 300 °C in subcritical acetic acid [28], of tetrachloro(trichloromethyl)benzene (unknown isomer/s) in a sealed vessel at 280 °C in subcritical water [29]. A later publication, reporting on the phytotoxicity activity of benzoic acid derivatives, includes production of a small library of 2,3,4,5-tetrachlorobenzamides; however, the synthetic details to this sub-family are scarce [30].

Here, decarbonylation occurred during the reaction of boron-bearing amines with phthalic anhydride to produce the corresponding secondary amides (**ortho 5**, **meta 5** and **para 5**). The reaction mechanism is hypothesised to proceed through an acyl substitution reaction occurring at an anhydride carbonyl by the nitrogen atom of the amine reagent. This is followed by decarbonylation of the adjacent carboxylate and formation of an anionic intermediate, with the resulting electron pair held in the aromatic C-sp^2^ orbital bearing the carboxylate group. This lone pair is thought to be stabilised by inductive and resonance effects via the nearby four electronegative chlorine atoms. This lone pair then strips off a proton intramolecularly from the quaternary nearby nitrogen atom, thus, providing the 2,3,4,5-tetrachlorobenzamide products.

### 2.3. Decarboxylation Reaction to Benzamides

Our methodology achieves the transformation of 2,3,4,5-tetrachlorophthalic anhydride and 2,3,4,5-tetrachlorophthalimide to the corresponding decarbonylated secondary amides in one synthetic step, avoiding the use of corrosive reagents and harsh conditions, and at lower temperature by heating the reaction mixture to 100 °C in DMF for 48 h to give the products **ortho 8** and **para 8** in moderate yields.

## 3. X-ray Crystallography Commentary

The structure of compound **meta 5** is shown in Figure 2. The asymmetric unit comprises two molecules, which exhibit essentially the same conformations and only one of these is shown. As expected, the trans-amide group is essentially planar (O1a-C-N1a-H 172.4°), while the two phenyl substituents are twisted (C_Ar_-C_Ar_-C-O1a 75.5°; C_Ar_-C_Ar_-N1a-C 33.1°) to minimise repulsion with the amide functional group.

In the structure of **ortho 8** (Figure 3)**,** the substituents on the B-substituted ring are in ortho positions. In comparison with **meta 5**, the insertion of a methylene group between the amide and B-substituted phenyl ring relieves torsional strain (C_Ar_-C_Ar_-C(H_2_)-N1 4.6°). The other structural features resemble those found in **meta 5,** defined by the dihedral angles O1-C-N1-H 173.8° and C_Ar_-C_Ar_-C-O1 76.6°, C_Ar_-C-N1a-C 61.1°.

The boronic ester groups in both structures are close to coplanar with the adjacent phenyl ring (out of plane twist <10°) and the C-B bonds (**ortho 8** 1.561(6) Å; **meta 5** 1.558(6) and 1.557(6) Å) are reinforced due to π-bonding with the sp^2^-hybridised B-atom. In twisted (purely σ-bonded) aromatic boronate esters (C_Ar_-C_Ar_-B-O ~90°), the C_Ar_-B bond is typically in a range 1.57–1.59 Å [31,32].

Intermolecular H-bonding in both structures comprises one-dimensional N-H…O chains of the amide functional group (in its trans H-N-C=O conformation). There are differences in the symmetry of the chains in the two structures. When the boronate ester is *ortho* on the benzamide ring, adjacent molecules are related by the *c* glide place (Figure 4A), orthogonal to the place of the page and propagating right to left, leading to an alternating (zig-zag) array of H-bonds along the chain.

In the structure of the **meta 5** isomer (Figure 4B), the presence of two molecules in the asymmetric unit (molecules A and B) breaks any symmetry relationship between adjacent molecules within the H-bonded chain, and the orientations of the N-H…O bonds are approximately the same along the chain, with rotation of the nearby aromatic rings facilitating closer packing.

## 4. Glycosidase Assay

In our laboratory we are interested in glycosidase modulation [13,18,19,21]. Screening for selectivity, as well as potency, is of paramount importance in carbohydrate-active enzyme research.

Our drugs and controls are, therefore, screened against two panels of glycosidases, respectively, in methanol (Table 1) and water (Table 2). This allows identification of the glycosidase-related disease area(s), selectivity profile and potency of biological action for each drug.

Following the glycosidase inhibition range recommendations [21], an IC_50_ value > 250 μm denotes weak inhibition, 100–249 μm denotes moderate inhibition, 10–99 μm good inhibition, 0.1–9 μm potent inhibition and <0.1 μm very potent inhibition. Our 2,3,4,5-tetrachlorobenzamides enter as a novel family of glycosidase inhibitors.

### 4.1. Glycosidases

The glycosidases screened are the following: rice α-glucosidase, yeast α-glucosidase, *Bacillus* α-glucosidase, rat intestinal maltase α-glucosidase, almond β-glucosidase, bovine liver β-glucosidase, coffee beans α-galactosidase, bovine liver β-galactosidase, Jack bean α-mannosidase, snail β-mannosidase, *Penicillium decumbens* α-L-rhamnosidase, bovine kidney α-L-fucosidase, *Eschierichia coli* β-glucuronidase, bovine liver β-glucuronidase, porcine kidney trehalase, *Aspergillus niger* amyloglucosidase and bovine kidney *N*-acetyl-β-glucosaminidase.

The disease areas for each glycosidase follow:

α-Glucosidase (EC 3.2.1.20) inhibition is linked to the management of diabetes, certain forms of hyperlipoproteinemia and obesity [33,34]. α-Glucosidase modulators also have potential as broad-spectrum anti-viral agents [35,36,37,38], for cancer [39] and lysosomal storage disorder Pompe disease [40,41].

β-Glucosidase (EC 3.2.1.21), α-Galactosidase (EC 3.2.1.22), and β-galactosidase (EC 3.2.1.23) are, respectively, linked to lysosomal storage disorders, such as Gaucher disease [42,43], Fabry disease [40], and GM1 gangliosidosis and Morquio syndrome B [40,44].

Other glycosidases involved in lysosomal storage disorders include α-mannosidase (EC 3.2.1.24), β-mannosidase (EC 3.2.1.25), α-L-fucosidase (EC 3.2.1.51), trehalase (EC 3.2.1.28), and β-glucuronidase (EC 3.2.1.31), whose malfunction cause, respectively, mannosidoses [45,46,47], fucosidosis [48,49], trehalase deficiency [50,51] and Sly disease [52]. Inhibition of β-glucuronidase may also help in controlling cancers and other diseases [53].

α-L-Rhamnosidase (EC 3.2.1.40) inhibition is linked to bacterial virulence [54,55]. Amyloglucosidase (EC 3.2.1.3) inhibition is important in diabetes management [56] and *N*-acetyl-β-D-glucosaminidase (EC 3.2.1.96) is involved in cancer progression and diabetic kidney disease [57].

### 4.2. Biological Activities for Phthalimides and Benzamides in the Literature

An overview of the literature in the field provides the following studies for phthalamides, 2,3,4,5-chlorophthalimides, and benzamides.

#### 4.2.1. Phthalamides

A siastatin-derivatised phthalimide produced a very potent inhibition of bovine kidney α-L-fucosidase (IC_50_ 0.013 μM) [58].

SAR studies on phthalimide analogues with *Saccharomyces cerevisiae* (yeast) α-glucosidase highlight good inhibitions for *N*-phenylphthalimides derivatised at the *ortho*-position with non-polar groups [59]. This was not found by us with our phthalimide drugs, but with our benzamide drugs.

On the other side of the molecule, substitutions of the phthalimide scaffold H atoms with other groups, such as amine or hydroxyl, tend to largely abrogate potency, but the introduction of nitro or alkyl groups tend to produce good inhibitors.

Another study on a library of *N*-phenylphthalimide derivatives showed the strongest potency against the three α-glucosidases screened was displayed by *N*-(2,4-dinitrophenyl)phthalimide, having two nitro groups in *ortho* and *para* positions. This is a moderate inhibitor of yeast α-glucosidase (IC_50_ 158 μM) and a good inhibitor of maltase (IC_50_ 51 μM), displaying no inhibition of sucrase [60].

*N*-Phenylphthalimide derivatives substituted with non-polar groups departing from the *ortho* position of the phenyl group keep providing inhibition of *Saccharomyces cerevisiae* α-glucosidase as low as 16 μM [61].

Other examples include the Hashimoto papers [62,63,64].

Several members of a phthalimide moiety connected by an alkyl chain to variously substituted phenoxy rings were screened against α-glucosidase. The inhibition potency appeared to be governed by the chain length of the substrate. Substrates possessing 10 carbons afforded the highest levels of activity, which were one to two orders of magnitude more potent than the known inhibitor 1-DNJ [4,65].

Bian and coworkers screened, against α-glucosidase, a series of *N*-substituted-(*p*-toluenesulfonylamino)phthalimides. Many analogues provide good inhibitions of the enzymes, with aromatic pendants and tethers containing 1–3 atoms generally producing the most potent inhibitions [3].

Another study of *N*-phenoxy-substituted phthalimides showed that the presence of a thiazolidine-2,4-dione or a rhodanine group, located at the 4-position of the phenyl ring, resulted in the best activity, with IC_50_ values as low as 5 μM against *Saccharomyces cerevisiae* α-glucosidase [66].

A series of phthalimide-benzamide-1,2,3-triazole hybrids showed good/moderate inhibitory activity against *Saccharomyces cerevisiae* α-glucosidase. The most potent compound displayed an IC_50_ of 40 μm [67].

##### 4.2.2. 2,3,4,5-Tetrachlorophthalimides

A 2,3,4,5-tetrachlorophthalimide, derivatised at the N-atom with a doubly *ortho*-substituted phenyl group, produced potent inhibitions. The use of linear alkyl chains departing from the phthalimide N atom produces potent and very potent inhibitions as the chain lengthens [59].

2,3,4,5-Tetrachlorophthalimides *N*-derivatised with a phenyl group attached directly to the N or through a linear alkyl tether (1–6 CH_2_ units) all displayed potent IC_50_ (3–11 μM) towards one α-glucosidase screened and more potent than the \1-DNJ control. The replacement of the four chlorine atoms with hydrogen atoms and the replacement of hydrogen atoms with other groups (e.g., nitro, amine) partially or completely abrogated the inhibitory activity of the drugs [6].

The same group also investigated other groups bonded to the phthalimide N, namely branched and cyclical alkyl groups and a dodecaborane group. All drugs displayed comparable or more potent activity (1–49 μM) than 1-DNJ. Cyclical alkyl groups and the borane group produced the most potent inhibitions [5].

The pendant groups attached to the phthalimide unit can clearly interact effectively with a number of sites in the vicinity of the active site, which is presumably occupied by the 2,3,4,5-tetrachlorophthalmide scaffold. This is highlighted by the variety and length of pendant groups. Hydrophobicity seems to be the common motif.

*N*-Phenyl-2,3,4,5-tetrachlorophthalimide derivatives substituted with non-polar groups departing from the *ortho*, *meta* and *para* positions of the phenyl group keep providing inhibition of α-glucosidase as low as 13 μM against *Saccharomyces*
*cerevisiae* [61].

##### 4.2.3. Benzamides

A series of *N*-substituted 1-aminomethyl-β-D-glucopyranoside derivatives was screened against *Saccharomyces cerevisiae* α-glucosidase, rat intestinal maltase α-glucosidase and sucrase. The most potent inhibitions were produced when the benzamide aromatic ring displayed groups in the *para* position to the amide. The three most potent compounds comprised *O*-acetyl groups in the 3-, 4- and 5-positions (IC_50_ 7.7 and 15.6 μM against rat intestinal maltase α-glucosidase and sucrase), a nitro group in the 4-position (IC_50_ 36.2 μM against *Saccharomyces cerevisiae* α-glucosidase) and an *O*-acetyl group in the 4-position (IC_50_ 96.5 μM against *Saccharomyces cerevisiae* α-glucosidase) [68].

1-DNJ derivatised with benzamides also produced potent to very potent inhibitions of sweet almond and *A. faecalis* β-glucosidases (IC_50_ 0.15–21 μM) [69].

Further, 24 Metronidazole-tethered benzamide triazoles demonstrated weak to good activity against β-glucuronidase, with no toxicity against 3T3 mouse fibroblast cell lines [70]. The most active compound has an IC_50_ 12.4 μM with an activity ~four-times higher than the standard inhibitor, D-saccharic acid-1,4-lactone (IC_50_ 45.8 μM) [53].

Iminosugar-benzamide derivatives displayed good to moderate inhibitions of *Aspergillus niger* amyloglucosidase, *Saccharomyces cerevisiae* α-glucosidase and human lymphocytes lysosomal α-glucosidase [71].

### 4.3. Biological Activities for Our Drug Library Screened in Methanol

Biological activities for controls, and our 2,3,4,5-tetrachloro phthalimides and benzamides, follow in Table 1 and Table 2.

For Table 1:

#### 4.3.1. Controls

Borocaptate sodium (BSH) and 4-borono-L-phenylalanine (BPA), and their ^10^B-enriched congeners ^10^B-BSHand ^10^B-BPA, were the controls. To our knowledge, these drugs have never been reported in a glycosidase assay. BSH and BPA are the drugs currently clinically used in BNCT. It is possible to see that none of them significantly inhibit any of the glycosidases in the panel at 100 or 1000 μM. In the panel, percent inhibitions range from a minimum value of 0 to a maximum value of 19.6.

#### 4.3.2. 2,3,4,5-Tetrachlorophthalimides

The 2,3,4,5-tetrachlorophthalimide drugs presented in this work do not provide any appreciable degree of inhibition, most likely because the spatial geometry and length of tether extending from the phthalimide scaffold are probably not able to reach the sites of interaction.

**para 3** and **meta 3** possess the CH_2_ spacer between the phthalimide and the aromatic boron group, with **para 3** displaying the boronate ester group in para position and **meta 3** in meta position to the phthalimide.

Of the benzamides, **para 5**, with no CH_2_ spacer between the aromatic boron group and the benzamide, shows moderate inhibition towards maltase α-glucosidase, with an IC_50_ 188 μM. Weak inhibitions are displayed towards bovine liver β-glucosidase (IC_50_ 543 μM) and bovine liver β-galactosidase (IC_50_ 333 μM).

**meta 5**, with no CH_2_ spacer between the aromatic boron group and the benzamide, again, shows moderate inhibition towards bovine liver β-glucosidase (IC_50_ 175 μM) and bovine liver β-galactosidase (IC_50_ 213 μM). Weak inhibition is observed towards maltase α-glucosidase, with an IC_50_ of 274 μM, and *E. coli* β-glucuronidase (IC_50_ 932 μM).

**para 5** and **meta 5** display the boronate ester group, respectively, in *para* and *meta* position from the benzamide. Both display partially selective inhibition profiles, inhibiting only 3–4 within the panel of 16 enzymes. Interestingly, their inhibitory profiles show a swap in potency, with **para 5** preferentially selecting one α-glucosidase and **meta 5** selecting β-glycosidases.

**ortho 5**, the *ortho* congener to **para 5** and **meta 5**, displays no significant inhibition of any of the enzymes. Hence, the location of the boronate ester group has a negative effect on drug–enzyme interactions, presumably either by preventing the drug from sitting in the active site as **para 5** and **meta 5** and/or abrogating any further interactions the aromatic boronate group may have with the enzyme.

#### 4.3.3. 2,3,4,5-Tetrachlorobenzamides

Benzamides **para 8** and **ortho 8**, possessing the CH_2_ spacer between the aromatic boron group and the benzamide, show partially selective inhibition of the same enzymes. **para 8** has the boronate ester group para to the benzamide, whereas **ortho 8** displays the boronate ester group ortho to the benzamide. Presumably, the lack of one of the carbonyl groups allows the pendant group to reach sites of favourable interactions with the enzymes. Since the enzymes inhibited are the same, it is surmised that the drug SAR profiles are similar.

**para 8** elicits moderate inhibition of maltase α-glucosidase, with an IC_50_ of 207 μM. **ortho 8** inhibits the same enzyme weakly, with a similar value of 305 μM. Weak inhibitions are displayed towards bovine liver β-glucosidase (IC_50_ = 702 and 922 μM, respectively).

The two main differences are seen:(a)In the inhibition of bovine liver β-galactosidase, which is inhibited weakly by **ortho 8** (IC_50_ 278 μM), but **para 8** inhibits the same enzyme with a good IC_50_ 74.7 μM. This is the most potent drug in the small libraries reported in this communication.(b)In the inhibition of almond β-glucosidase by **ortho 8**, which is on the edge of moderate/weak, with an IC_50_ of 254 μM.

The drugs that show appreciable inhibition all inhibit the same glycosidase enzymes (maltase α-glucosidase, bovine liver β-glucosidase and bovine liver β-galactosidase). Furthermore, all drugs selectively inhibit glycosidases of animal origin vs. glycosidases of plant or bacterial origin within the same glycosidase class. This is a positive result for applications in a human disease medicinal chemistry context.

### 4.4. Biological Activities for Our Drug Library Screened in Water

For Table 2:

Some differences were noted in the results obtained from this second laboratory that used just water to suspend or dissolve the compounds. The compounds did not fully go into solution but nonetheless, activities were observed and so they are included here for comparison.

No appreciable inhibition was detected for any of the controls and compounds against *Bacillus* α-glucosidase, Jack bean α-glucosidase, bovine kidney A-acetyl-β-glucosaminidase and bovine liver β-glucuronidase.

Three potent % inhibitions at a concentration of ~400 μM were seen for **meta 5** (82.8%) and **para 8** (99.5%) against yeast α-glucosidase, and for **ortho 8** (83.4%) against almond β-glucosidase.

## 5. Cancer Assay and Structure Activity Relationships

In our laboratory, we are interested in BNCT as a potentially broad-spectrum approach to cancer management. It would be advantageous upon irradiation, if the boron-containing drugs accumulate more selectively in cancer cells vs. healthy cells [72,73]. In case the drugs do not accumulate selectively in cancer vs. healthy cells, the delivery of radiation is required with greater precision.

BNCT is essentially a non-invasive radiation technique and the least destructive currently available [23,24,25]. Use of a borylated drug in BNCT would ideally require that it is non-toxic in the absence of radiation. Following a first study of synthesis, purification and toxicity of organic-boron-containing drugs for BNCT applications [13], we report here, two further families of potential BNCT agents.

BNCT agents that contain organic boron groups are preferable to ones containing inorganic boron. The currently utilised sodium borocaptate, BSH, with its inorganic boron atoms, raises several toxicity concerns [74,75]. Boronophenylalanine, BPA, which contains the organic boronic acid moiety, has long been known to show no discernible toxicity [76]. Similarly, in this area, candidate BNCT agents containing an organic boron group should be more likely to reach the clinic.

It has long been known that organic boron is an essential element for plants [77,78] and is likely to be essential for human and animal health [79].

When comparing toxicological data for organic boron-containing molecules with their non-borylated congeners, the trend is that the presence of organic boron lowers toxicity profiles. For example, benzene has an LD_50_ (lethal dose) of 125 mg/kg (human, oral) [80] and an LCLO (lethal concentration) of 20,000 ppm (human, 5 min); it is carcinogenic, and also possibly mutagenic. The NIOSH Permissible Exposure Limit for benzene is 1 ppm, the Recommended Exposure Limit is 0.1 ppm, and the Immediately Dangerous to Life and Health concentration is at 500 ppm [81].

On the other hand, phenylboronic acid has an LD_50_ 740 mg/kg (rat, oral) [82], with no entry for RTECS, ACGIH, IARC, or NTP.

If a BNCT agent also has growth inhibition capability against cancer cells, then it is important to screen them in more complex biological systems, such as spheroids, as we did recently [72,73].

Table 3 shows, on a blue background, the percentage cell growth inhibition in response to 25 μM of the drug. In this case, a higher value correlates with a greater growth inhibition. Inhibition value ranges have been colour coded, according to potency, with 80–100% inhibition in red, 60–79% in orange, 35–59% in green, 10–34% in blue and 0–9% in black.

On the green background, the GI_50_ values are provided for the most potent drugs. The GI_50_ value provides the concentration in μM that induces a 50% cell growth inhibition. In this case, a lower value correlates with greater growth inhibition.

### 5.1. Analysis of the Percent Cell Growth Inhibition Data

BSH, BPA and their ^10^B-enriched congeners ^10^B-BSH and ^10^B-BPA are the controls. BSH and BPA are the drugs currently clinically used in BNCT. It is possible to see that none of them significantly inhibit cell growth at 25 μM. Percent cell growth inhibitions range from <0% to 19% in our panel of 10 cancer cell lines (HT29, U87, MCF-7, A2780, H460, A431, Du145, BE2-C, SJ-G2 and MIA-Pa-Ca2) and a normal cell line (MCF10A). Specifically, only 10 out of 44 entries for the controls were double-digit (10% or greater) percent inhibitions. The other 34 entries contained inhibition values between 0.01% and 9.99%, and nine entries with values of 0% or <0%.

It is evident that more efficacious BNCT agents are required.

When analysing the data for the borylated drugs, a number of Structure–Activity Relationship considerations can be evinced.

A general overarching consideration is that the vast majority of percent cell growth inhibitions for the borylated drugs are significantly greater than the percent cell growth inhibitions for the BSH and BPA controls.

There are three drugs that possess potent inhibitions, namely tetrachlorophthalimides **para 3** and **meta 3**, and tetrachlorobenzamide **ortho 5**.

**para 3** is the only tetrachlorophthalimide that displays potent inhibition. It displays the boronate ester group in *para* position to the phthalimide and it has the CH_2_ spacer between the phthalimide and the aromatic boron group. Inhibitions range from 98% to >100% for 9 out of 10 cancer cell lines and for the normal cell line. Only the A431 cancer cell line displays a lower inhibition (85%).

Tetrachlorophthalimide **meta 3** possesses a similar structure to **para 3,** with the boronate ester group in *meta* position to the phthalimide. The installation of the boronate ester group in the *meta* position reduces potency significantly for all cancer lines and the normal cell line, though to varying extents. The smallest reduction in inhibition is seen in cell lines U87 (70%), MCF-7 (67%), A2780 (70%) and MCF10A (69%). A further loss in growth inhibition is seen in HT29 (53%), H460 (42%), A431 (48%), Du145 (43%) and BE2-C (45%). The greatest loss of growth inhibition is displayed in cell lines SJ-G2 (35%) and MIA-Pa-Ca2 (29%). Hence, the location of the boron ester group in *para* position greatly favours cell growth inhibition. These two drugs likely interact in similar ways with the cells, with the boronate ester group likely trying to interact with the same site/s (designated Site A for discussion purposes), but not managing quite as effectively when it is in the *meta* position.

Tetrachlorobenzamide **ortho 5**, displaying the boronate ester group in *ortho* position and having no CH_2_ spacer between the benzamide and the aromatic boron group, also displays potent cell growth inhibition and a significant level of cell selectivity. This capability makes this drug the most interesting from a medicinal chemistry perspective. Cell selectivity (in particular, cancer versus healthy cell selectivity) is an area of active research in our group [72]. The removal of one of the carbonyl groups allows for greater conformational flexibility to this molecule, which may allow the boronate ester to interact with a different site than **para 3** and **meta 3** (designated Site B for discussion purposes). The *ortho*-phthalimide congener has not been synthetically achievable so far, and so it was not tested. Percent cell growth inhibitions range from 97 to >100 for most cancer cell lines (HT29, MCF-7, A2780, H460, A431, Du145, BE2-C and MIA-Pa-Ca2). It is 92% for SJ-G2 and drops dramatically for U87 (58%) and for the normal cell line MCF10A (58%). This selectivity between cancer versus healthy cells is a highly desirable drug capability.

The comparison in percent inhibition between benzamide **ortho 5** and its congener **ortho 8** is particularly interesting. The only structural difference is the CH_2_ spacer between the benzamide and the aromatic boron group. However, there is a complete abrogation of cell growth inhibition produced by **meta 3**. It can be evinced that **ortho 8** likely interacts with the same sites **para 3** and **meta 3** interact with, whereas **ortho 5** has a different mode of action, interacting with another site that seems to be overexpressed in all cancer cell lines, apart from U87 and the normal cancer cell line MCF10A.

Of the drugs that inhibited cell growth to a lesser extent, **para 8** is the benzamide analogue of **para 3**. **para 8** is thought to also interact with Site A due to structural similarities with **para 3**; however, it shows a significantly reduced growth inhibition, probably due to the boronate ester group interacting not as efficiently on Site A. In this case as well, cell selectivity is displayed in inhibition. Normal cells MCF10A (2%) and SJ-G2 (8%), Du145 (<0%) and A431 (<0%) were not inhibited, whereas BE2-C (15%), MCF-7 (18%) and U87 (14%) were minimally inhibited, MIA-Pa-Ca2 (23%), H460 (21%) and HT29 (36%) were somewhat more inhibited, and finally, A2780 was significantly inhibited (58%).

Benzamide **para 5**, not possessing the CH_2_ spacer between the benzamide and the *para*-aromatic boron group, is thought to interact with Site A, due to structural similarities with its phthalimide congeners; however, it does not efficiently interact with Site A. This may be due to the boronate ester not reaching Site A due to the lack of the CH_2_ spacer and the greater degree of conformational flexibility deriving from the removal of the carbonyl group from the phthalimide scaffold. This results in an overall reduction in cell growth inhibition. In this case as well, cell selectivity is displayed in inhibition. Normal cells MCF10A (2%) and Du145 (1%) were not inhibited, whereas U87 (21%) and SJ-G2 (28%) were somewhat more inhibited, MCF-7 (42%), A2780 (50%), H460 (54%), BE2-C (54%) and MIA-Pa-Ca2 (52%) were significantly inhibited, and finally, HT29 (69%) and A431 (64%) were inhibited the most.

Benzamide **meta 5**, not possessing the CH_2_ spacer between the benzamide and the aromatic boron group, shows selective and significant inhibition for A2780 (49%). The benzamide structure provides a greater degree of conformational flexibility, likely placing the *meta*-positioned boronate ester somewhere in between Site A and Site B, and preventing it from interacting efficiently with either.

Benzamide **ortho 8**, possessing the CH_2_ spacer between the benzamide and the aromatic boron group, is thought to interact with Site A, due to structural similarities with its phthalimide congeners; however, it does not efficiently interact with Site A, thus, showing almost complete abrogation of cell growth inhibition.

Based on Structure–Activity Relationship data obtained, two modes of cell growth inhibition are put forward, Mode of Action A, which arises from drugs interacting at Site A, and Mode of Action B, arising from drug **ortho 5** interacting at Site B. Both Modes of Action can be elicited in selective ways by drugs **para 8** (for Mode of Action A) and **ortho 5** (for Mode of Action B) and displaying minimal or zero inhibition on the normal cell line.

### 5.2. Analysis of the GI_50_ Data

The GI_50_ was measured for the three most potent drugs, tetrachlorophthalimides **para 3** and **meta 3**, and tetrachlorobenzamide **ortho 5**. **para 3** shows consistent potency via GI_50_ values between 3 and 18 μM for all cancer cell lines and the normal cell line. Similarly, **meta 3** displays consistently potent GI_50_ values between 18 and 38 μM for all cancer cell lines and the normal cell line. **ortho 5** displays consistently potent GI_50_ values between 11 and 27 μM for all cancer cell lines and the normal cell line.

## 6. Experimental

### 6.1. Glycosidase Inhibition Experimental from Laboratory 1

In Table 1

The enzymes α-glucosidase (from yeast), β-glucosidases (from almond and bovine liver), α-galactosidase (from coffee beans), β-galactosidase (from bovine liver), α-mannosidase (from Jack bean), β-mannosidase (from snail), α-L-rhamnosidase (from *Penicillium decumbens*), α-L-fucosidase (from bovine kidney), trehalase (from porcine kidney), β-glucuronidases (from *E. coli* and bovine liver), amyloglucosidase (from *A. niger*), *para*-nitrophenyl glycosides, and various disaccharides were purchased from Sigma-Aldrich Co (St Louis, MO, USA).

Brush border membranes were prepared from the rat small intestine according to the method of Kessler et al. [83] and were assayed at pH 6.8 for rat intestinal maltase using maltose. For rat intestinal maltase, porcine kidney trehalase, and *A. niger* amyloglucosidase activities, the reaction mixture contained 25 mM maltose and the appropriate amount of enzyme, and the incubations were performed for 10–30 min at 37 °C. The reaction was stopped by heating at 100 °C for 3 min. After centrifugation (600× *g*; 10 min), the resulting reaction mixture was added to the Glucose CII-test Wako (Wako Pure Chemical Ind., Osaka, Japan). The absorbance at 505 nm was measured to determine the amount of the released D-glucose. Other glycosidase activities were determined using an appropriate *para*-nitrophenyl glycoside as substrate at the optimum pH of each enzyme. The reaction mixture contained 2 mM of the substrate and the appropriate amount of enzyme. The reaction was stopped by addition of 400 mM Na_2_CO_3_. The released *para*-nitrophenol was measured spectrometrically at 400 nm. All reactions run in methanol.

### 6.2. Glycosidase Inhibition Experimental from Laboratory 2

In Table 2

All enzymes and *para*-nitrophenyl substrates were purchased from Sigma. Enzymes were assayed at 27 °C in 0.1 M citric acid/0.2 M disodium hydrogen phosphate buffers at the optimum pH for the enzyme. The incubation mixture consisted of 10 μL enzyme solution, 10 μL of 1 mg/mL aqueous solution of extract and 50 μL of the appropriate 5 mM *para*-nitrophenyl substrate made up in buffer at the optimum pH for the enzyme. The reactions were stopped by addition of 70 μL 0.4 M glycine (pH 10.4) during the exponential phase of the reaction, which had been determined at the beginning using uninhibited assays in which water replaced inhibitor. Final absorbances were read at 405 nm using a Versamax microplate reader (Molecular Devices). Assays were carried out in triplicate, and the values given are means of the three replicates per assay. All reactions run in water.

### 6.3. Cancer Screening Experimental

All test agents were prepared as stock solutions (20 mM) in dimethyl sulfoxide (DMSO) and stored at −20 °C. Cell lines used in the study included HT29 (colorectal carcinoma); U87, SJ-G2, (glioblastoma); MCF-7, (breast carcinoma); A2780 (ovarian carcinoma); H460 (lung carcinoma); A431 (skin carcinoma); Du145 (prostate carcinoma); BE2-C (neuroblastoma); MiaPaCa-2 (pancreatic carcinoma); and SMA560 (spontaneous murine astrocytoma), together with the one non-tumour-derived normal breast cell line (MCF10A). All cell lines were incubated in a humidified atmosphere 5% CO_2_ at 37 °C. The cancer cell lines were maintained in Dulbecco’s modified Eagle’s medium (DMEM; Sigma, Australia) supplemented with foetal bovine serum (10%), sodium pyruvate (10 mM), penicillin (100 IUmL^−1^), streptomycin (100 µg mL^−1^) and L-glutamine (2 mM).

The non-cancer MCF10A cell line was maintained in DMEM:F12 (1:1) cell culture media, 5% heat-inactivated horse serum, supplemented with penicillin (50 IUmL^−1^), streptomycin (50 µg mL^−1^), HEPES (20 mM), L-glutamine (2 mM), epidermal growth factor (20 ng mL^−1^), hydrocortisone (500 ng mL^−1^), cholera toxin (100 ng mL^−1^) and insulin (10 mg mL^−1^).

Growth inhibition was determined by plating cells in duplicate in medium (100 µL) at a density of 2500–4000 cells per well in 96-well plates. On day 0 (24 h after plating), when the cells were in logarithmic growth, medium (100 µL) with or without the test agent was added to each well. After 72 h drug exposure, growth inhibitory effects were evaluated using the MTT (3-(4,5-dimethyltiazol-2-yl)-2,5-diphenyltetrazolium bromide) assay and absorbance read at 540 nm. The percentage growth inhibition was calculated at a fixed concentration of 25 µM, based on the difference between the optical density values on day 0 and those at the end of drug exposure. Each data point is the mean ± the standard error of the mean (SEM) calculated from three replicates which were performed on separate occasions and separate cell line passages.

### 6.4. Chemistry Experimental

#### 6.4.1. General Experimental

Reaction solvents were purchased from the Aldrich Chemical Company (St Louis, MO, USA) in sure-seal^TM^ reagent bottles. All other solvents (analytical or HPLC grade) were used as supplied without further purification. Deuterated chloroform (CDCl_3_) and water (D_2_O) were used as NMR solvent. Triethylamine, sodium hydride (60% dispersion in mineral oil), tetrachlorophthalimide and tetrachlorophthalic anhydride were purchased from Sigma Aldrich. All boron-containing reagents were purchased from Boron Molecular, apart from BSH (>97%), ^10^B-BSH (>97%), BPA (>98%) and ^10^B-BPA (>98%) which came from Katchem spol. s r. o. The reagents were used as provided without further purification, with NMR analysis confirming an acceptable degree of purity and correct structural identity.

Purification via silica gel column chromatography was performed on Davisil 40–63-micron silica gel.

Thin layer chromatography (t.l.c.) was performed on aluminium sheets coated with 60 F254 silica by Merck and visualised using UVG-11 Compact UV lamp (254 nm) or stained with the cerium molybdate stain (12.0 g ammonium molybdate, 0.5 g ceric ammonium molybdate in 15 mL concentrated sulfuric acid and 235 mL distilled water).

Nuclear Magnetic Resonance (NMR) spectra were recorded on Bruker Ascend^TM^ 400 in deuterated chloroform (CDCl_3_). Chemical shifts (δ) are quoted in ppm and coupling constants (*J*) in Hz. Residual signals from the CDCl_3_ (7.26 ppm for ^1^H-NMR and 77.16 ppm for ^13^C-NMR) were used as an internal reference [84].

Infrared spectroscopy (IR) spectra were obtained on a PerkinElmer Spectrum Two Spectrometer and on a PerkinElmer Spectrum 2 with UATR. Only characteristic peaks are quoted and in units of cm^−1^.

High-resolution mass spectrometry (HRMS) spectra were obtained from samples suspended in acetonitrile (1 mL with 0.1% formic acid at a concentration of ~1 mg/mL, before being further diluted to ~10 ng/μL in 50% acetonitrile/water containing 0.1% formic acid). Samples were infused directly into the HESI source of a Thermo Scientific Q Exactive™ Plus Hybrid Quadrupole-Orbitrap™ Mass Spectrometer using an on-board syringe pump at 5 μL/min. Data were acquired on the QE+ in both positive and negative ion mode at a target resolution of 70,000 at 200 *m*/*z*. The predominant ions were manually selected for MS/MS fragmentation (collision energies were altered for each compound to obtain sufficient fragmentation). Data analysis of each sample was performed manually using Thermo Qualbrowser whilst the Isotopic Patterns of predicted chemical formula were modelled using Bruker Compass Isotope Pattern.

Crystallographic data were collected on an Oxford Diffraction Gemini CCD diffractometer employing either graphite-monochromated Mo-Kα radiation (0.71073 Å) or Cu-Kα (1.54184 Å). The sample was cooled to 190 K with and Oxford Cryosystems Desktop Cooler. Data reduction and empirical absorption corrections were performed with Oxford Diffraction CrysAlisPro software. Structures were solved by direct methods and refined with SHELXL [85]. All non-H atoms were refined with anisotropic thermal parameters. The crystal of **ortho 8** was a non-merohedral twin which was refined using the HKLF 5 mode in SHELX. Molecular structure diagrams were produced with Mercury [86]. The data in CIF format were deposited at the Cambridge Crystallographic Data Centre (CCDC 215189 and 2151899).

Melting points were taken on a Dynalon SMP100 Digital Melting Point Device and are uncorrected.

#### 6.4.2. Experimental

##### From Synthetic Strategy 1


**4,5,6,7-Tetrachloro-2-(3-(4,4,5,5-tetramethyl-1,3,2-dioxaborolan-2-yl)benzyl)isoindoline-1,3-dione meta 3**




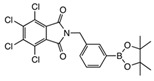



Tetrachlorophthalic anhydride **1** (445 mg, 1.558 mmol, 1.2 equiv.) and 3-(aminomethyl)phenylboronic acid pinacol ester hydrochloride **meta 2** (353 mg, 1.309 mmol, 1.0 equiv.) were stirred in *N*,*N* dimethylformamide (8 mL) until dissolved. Triethylamine (328 mg, 0.45 mL, 3.245 mmol, 2.5 equiv.) was then added dropwise to reaction mixture. A white precipitate crashed out upon addition of triethylamine. The stirring reaction mixture was then heated to 85 °C. After 48 h the reaction mixture was gravity filtered to remove the precipitate (triethylamine salt and excess tetrachlorophthalic anhydride by NMR analysis), which was collected in a sample vial and retained for analysis. The filtrate was then evaporated and the solid recrystallized from methanol (20 mL) to afford the product **meta 3** as a pale creamy yellow solid (389 mg, 60.0%). M.p. 316–320 °C. *m*/*z* (HRMS ES^+^): Relative intensities for C_21_H_19_BCl_4_NO_4_, [M + H]^+^, found 499.01957 (22%), 500.01581 (82%), 501.01824 (40%), 502.01293 (100%), 503.04453 (34%), 504.01001 (45%), 505.04176 (16%), 506.00706 (9%); calculated 499.01921 (18%), 500.01596 (76%), 501.01748 (40%), 502.01323 (100%), 503.01528 (34%), 504.01064 (50%), 505.01292 (13%), 506.00829 (12%). ν_max_ (thin film, cm^−1^): 2979 (w, alkyl CHs), 1776 (w, C=O), 1716 (s, C=ON, amide I), 1606, 1486 (w, ArC=C), 1432 (m, C-B), 1389, 1361, 1330, 1075 (s, sp^2^ B-O). δ_H_ (CDCl_3_, 400 MHz): 7.85 (1 H, s, H^a^), 7.73 (1 H, d, *J*_Hb,Hc_ 7.6 Hz, H^b^), 7.52 (1 H, d, *J*_Hd,Hc_ 7.6 Hz, H^d^), 7.33 (1 H, t, *J*_Hc,Hb/Hd_ 7.6 Hz, H^c^), 4.85 (2 H, s, CH_2_), 1.34 (12 H, s, 4 × CH_3_). δ_c_ (CDCl_3_, 100 MHz): 163.2 (2 × C=O), 140.1 (Ar*C_q_*-CH_2_), 135.00, 134.6 (2 × ArCH), 134.5 (2 × Ar*C_q_*-CO), 131.7 (ArCH), 129.7, 128.2 (ArCH), 127.60 (4 × Ar*C_q_*-Cl), 83.9 (2 × pinacol C_q_), 42.4 (N-CH_2_) and 24.8 (4 × pinacol CH_3_). Ar*C_q_*-B is not visible. δ_B_ (dissolved in CDCl_3_, 96 MHz): 31.5 ppm (Appendix A).


**4,5,6,7-Tetrachloro-2-(4-(4,4,5,5-tetramethyl-1,3,2-dioxaborolan-2-yl)benzyl)isoindoline-1,3-dione para 3**




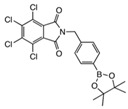



Tetrachlorophthalic anhydride **1** (453 mg, 1.558 mmol, 1.2 equiv.) and 4-(aminomethyl)phenylboronic acid pinacol ester hydrochloride **para 2** (356 mg, 1.321 mmol, 1.0 equiv.) were stirred in *N*,*N*-dimethylformamide (8 mL) until dissolved. Triethylamine (328 mg, 0.45 mL, 3.245 mmol, 2.5 equiv.) was then added dropwise to reaction mixture. A white precipitate crashed out upon addition of triethylamine. The stirring reaction mixture was then heated to 85 °C. After 48 h the reaction mixture was gravity filtered. The filtrate was then evaporated and the solid recrystallised from petroleum ether (60:80)/ethanol (30/80 mL) to afford the product **para 3** as a pale creamy yellow solid (289 mg, 44.4%). M.p. 242–246 °C. *m*/*z* (HRMS ES^+^): Relative intensities for C_21_H_19_BCl_4_NO_4_, [M + H]^+^, found 499.01957 (22%), 500.01581 (82%), 501.01824 (40%), 502.01293 (100%), 503.04453 (34%), 504.01001 (45%), 505.04176 (16%), 506.00706 (9%); calculated 499.01921 (18%), 500.01596 (76%), 501.01748 (40%), 502.01323 (100%), 503.01528 (34%), 504.01064 (50%), 505.01292 (13%), 506.00829 (12%). ν_max_ (thin film, cm^−1^): 2978 (m, ArCH_2_), 2938, 2893 (w, alkyl C-H), 1778 (w, C=O), 1715 (s, C=ON, amide I), 1512 (s, C=ON, amide II), 1433 (m, C-B), 1373, 1360, 1345, 1087 (s, sp^2^ B-O), 662 (s, C-B(-O)_2_, out of plane bending). δ_H_ (CDCl_3_, 400 MHz): 7.77 (2 H, d, *J* 8.0 Hz, 2 × ArH), 7.42 (2 H, d, *J* 8.0 Hz, 2 × ArH), 4.85 (2 H, s, CH_2_), 1.32 (12 H, s, 4 × CH_3_). δ_C_ (CDCl_3_, 100 MHz): 163.4 (C=O), 140.3 (Ar*C_q_*-CH_2_), 138.3 (2 × Ar*C_q_*-CO), 135.4 (2 × ArCH), 129.9, 127.7 (4 × Ar*C_q_*-Cl), 128.3 (2 × ArCH), 84.0 (2 × pinacol C_q_), 42.6 (N-CH_2_) and 25.0 (4 × pinacol CH_3_). Ar*C_q_*-B is not visible. δ_B_ (dissolved in CDCl_3_, 96 MHz): 31.1 ppm.


**2,3,4,5-Tetrachloro-N-(2-(4,4,5,5-tetramethyl-1,3,2-dioxaborolan-2-yl)phenyl)benzamide ortho 5**




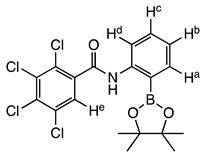



Tetrachlorophthalic anhydride (560 mg, 1.959 mmol, 1.2 equiv.) and 2-amino phenylboronic acid pinacol ester (352 mg, 1.607 mmol, 1.0 equiv.) were stirred in *N*,*N*-dimethylformamide (9 mL) to give a cloudy orange solution. Triethylamine (404 mg, 0.56 mL, 3.994 mmol, 2.5 equiv.) was added dropwise and stirred at 85 °C for 72 h, and at 105 °C for a further 72 h. Evaporation gave a dark brown oily residue. Recrystallisation attempts using ethanol (20 mL) and petroleum ether (60/80) (20 mL)/chloroform (5 mL) did not yield enough product. The residue was purified by flash column chromatography (hexane:acetone, 1:1). After evaporation a recrystallization using chloroform (20 mL) gave a filtrate which was left to stand for 3–4 days. Crystallisation gave product **ortho 5** (284 mg, 37.4%). M.p. 106–108 °C. *m*/*z* (HRMS ES^+^): Relative intensities for C_19_H_19_BCl_4_NO_3_, [M + H]^+^, found 459.02226 (26%), 460.02068 (77%), 461.02147 (35%), 462.01769 (100%), 463.02069 (28%), 464.01455 (46%), 465.01838 (11%), 466.01142 (10%); calculated 459.02429 (18%), 460.02101 (77%), 461.02249 (39%), 462.01824 (100%), 463.02026 (32%), 464.01558 (50%), 465.01788 (12%), 466.01314 (11%). ν_max_ (thin film, cm^−1^): 3357 (m, sh, NH, hydrogen-bonded), 2980, 2933 (m, alkyl CHs), 1690 (s, C=ONH, amide I), 1612 (w, ArC=C), 1580 (m, C=ONH, amide II), 1536 (w, C=ONH, bending), 1450 (m, C-N), 1407 (m, C-B), 1351, 1323, 1303, 1140 (s, sp^2^ B-O), 758 (s, C-Cl), 652 (s, C-B(-O)_2_, out of plane bending). δ_H_ (CDCl_3_, 400 MHz), major:minor conformer 2:1. Major conformer: 9.90–9.81 (1 H, broad s, NH), 8.59 (1 H, d, *J*_Ha,Hb_ 8.4 Hz, H^a^), 7.82 (1 H, dd, *J*_Hd,Hc_ 7.2 Hz, *J*_Hd,Hb_ 1.6 Hz, H^d^), 7.71 (1 H, s, H^e^), 7.53 (1 H, td, *J*_Hb,Ha/Hc_ 7.6 Hz, *J*_Hb,Hd_ 1.5 Hz, H^b^), 7.16 (1 H, td, *J*_Hc,Hd/Hb_ 7.5 Hz, Hz, *J*_Hc,Ha_ 0.9 Hz, H^c^), 1.33 (12 H, s, 4 × CH_3_). δc (CDCl_3_, 100 MHz): 162.3 (C=O), 144.0 (NHCq), 136.8 (Cq), 136.5 (C^d^), 134.9 (Cq), 133.2 (C^b^), 132.7, 130.2 (2 × Cq), 127.8 (C^e^), 124.1 (C^c^), 120.0 (C^a^), 84.8 (2 × pinacol Cq) and 25.0 (4 × pinacol CH_3_); note, Ar*C_q_*-B is not visible. δ_B_ (dissolved in CDCl_3_, 96 MHz): 31.1 ppm.


**2,3,4,5-tetrachloro-N-(3-(4,4,5,5-tetramethyl-1,3,2-dioxaborolan-2-yl)phenyl)benzamide meta 5**




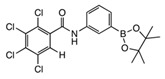



Tetrachlorophthalic anhydride (580 mg, 2.029 mmol, 1.2 equiv.) and 3-aminophenylboronic acid pinacol ester (354 mg, 1.616 mmol, 1.0 equiv.) were stirred in *N*,*N* dimethylformamide (9 mL) to give a golden-yellow solution. Triethylamine (404 mg, 0.56 mL, 3.994 mmol, 2.5 equiv.) was added dropwise and the reaction stirred at 85 °C for 48 h and then at 105 °C for a further 72 h. The reaction mixture was evaporated and recrystallised with ethanol (20 mL) and the reaction solution left to crystallise in the freezer for three days at −25 °C to produce **meta 5** as light-brown crystals (395 mg, 52%). M.p. 36–38 °C. *m*/*z* (HRMS ES^+^): Relative intensities for C_19_H_18_BCl_4_NO_3_, [M + H]^+^, found 459.02469 (19%), 460.02087 (80%), 461.02264 (35%), 462.01773 (100%), 463.02026 (30%), 464.01450 (47%), 465.01779 (11%), 466.01114 (9%); calculated 459.02429 (18%), 460.02101 (77%), 461.02249 (39%), 462.01824 (100%), 463.02026 (32%), 464.01558 (50%), 465.01788 (12%), 466.01314 (11%). δ_H_ (CDCl_3_, 400 MHz): 7.95 (1 H, dd, *J*_Hb,Hc_ 7.7 Hz, *J*_Hb,Ha_ 1.2 Hz, H^b^), 7.80 (1 H, s, NH), 7.76 (1 H, d, *J*_Ha,Hb_ 1.6 Hz, H^a^), 7.72 (1 H, s, H^e^), 7.63 (1 H, d, *J*_Hd,Hc_ 7.2 Hz, H^d^), 7.41 (1 H, t, *J*_Hc,Hb/Hd_ 7.6 Hz, H^c^) and 1.34 (12 H, s). δ_B_ (dissolved in CDCl_3_, 96 MHz): 31.7. X-ray Crystallographic analysis.


**2,3,4,5-tetrachloro-N-(4-(4,4,5,5-tetramethyl-1,3,2-dioxaborolan-2-yl)phenyl)benzamide para 5**




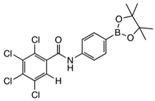



Tetrachlorophthalic anhydride (560 mg, 1.958 mmol, 1.2 equiv.) and 4-amino phenylboronic acid pinacol ester (353 mg, 1.611 mmol, 1.0 equiv.) were stirred in *N*,*N*-dimethylformamide (9 mL) into a cloudy orange solution. Sodium hydride (168 mg, 17.51 mmol, 2.5 equiv.) was then carefully added as hydrogen gas was given off. The reaction mixture was then heated to 100 °C for 72 h. After cooling, two drops of R.O. water were added to quench the reaction and evaporated. The residue was eluted in ethanol (10 mL) and a black solid precipitated. The filtrate contained product **para 5**, which was coevaporated with DCM (354 mg, 46%). M.p. 218–220 °C. *m*/*z* (HRMS ES^+^): Relative intensities for C_19_H_19_BCl_4_NO_3_, [M + H]^+^, found 459.02445 (20%), 460.02049 (77%), 461.02242 (38%), 462.01751 (100%), 463.02008 (30%), 464.01432 (47%), 465.01768 (12%), 466.01123 (10%); calculated 459.02429 (18%), 460.02101 (77%), 461.02249 (39%), 462.01824 (100%), 463.02026 (32%), 464.01558 (50%), 465.01788 (12%), 466.01314 (11%). ν_max_ (thin film, cm^−1^): 3266 (w, br, NH), 2976, 2926, 2855 (m, alkyl CHs), 1661 (s, C=ONH, amide I), 1597 (w, ArC=C), 1529 (m, C=ONH, amide II), 1505 (w, C=ONH, bending), 1444 (m, C-N), 1399 (m, C-B), 1389, 1356, 1316, 1084 (s, sp^2^ B-O), 833 (s, C-Cl), 653 (s, C-B(-O)_2_, out of plane bending). δ_H_ (CDCl_3_, 400 MHz): 7.83 (2 H, d, *J* 8.0 Hz, 2 × ArH), 7.75 (1 H, s, H^e^), 7.65 (1 H, s, NH), 7.61 (2 H, d, *J* 7.6 Hz, 2 × ArH), 1.35 (12 H, s, 4 × CH_3_). δc (CDCl_3_, 100 MHz): 136.2 (2 × ArCH), 128.7 (C^e^), 119.1 (2 × ArCH), 84.0 (2 × pinacol C_q_) and 25.0 (4 × pinacol CH_3_); note, other Cq are not visible. δ_B_ (dissolved in CDCl_3_, 96 MHz): 31.8 ppm.

##### From Synthetic Strategy 2


**2,3,4,5-tetrachloro-N-(2-(4,4,5,5-tetramethyl-1,3,2-dioxaborolan-2-yl)benzyl)benzamide ortho 8**




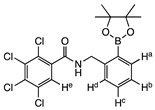



Tetrachlorophthalimide **6** (352 mg, 1.235 mmol, 1.0 equiv.) and 2-(bromomethyl)phenylboronic acid pinacol ester **ortho 7** (455 mg, 1.838 mmol, 1.2 equiv.) were stirred in *N*,*N*-dimethylformamide (9 mL). Sodium hydride (123 mg, 12.80 mmol, 2.5 equiv.) was then carefully added as hydrogen gas was given off to give a pale creamy yellow reaction mixture and heated to 100 °C. After 48 h the reaction mixture was cooled to room temperature and a yellow precipitate formed. Two drops of R.O. water were added to quench. Evaporation afforded a crude residue, which was then dissolved in chloroform (30 mL) and gravity filtered. The filtrate was dried affording **ortho 8** (371 mg, 60%). M.p. 120–122 °C. *m*/*z* (HRMS ES^+^): Relative intensities for C_21_H_19_BCl_4_NO_4_, [M + H]^+^, 473.04023 (20%), 474.03660 (80%), 475.03831 (39%), 476.03341 (100%), 477.03592 (32%), 478.03016 (46%), 479.03318 (12%), 480.02677 (9%); calculated 473.03994 (18%), 474.03668 (76%), 475.03818 (40%), 476.03392 (100%), 477.03596 (33%), 478.03129 (50%), 479.03358 (13%), 480.02889 (11%). δ_H_ (CDCl_3_, 400 MHz), major:minor conformers 5:1–3:1. Major conformer: 7.87 (1 H, dd, *J*_Ha,Hb_ 7.6, *J*_Ha,Hc_ 1.2 Hz, H^a^), 7.61 (1 H, s, H^e^), 7.49 (1 H, d, *J*_Hd,Hc_ 7.5 Hz, H^d^), 7.45 (1 H, td, *J*_Hc,Hd_ 7.5, *J*_Hc,Ha_ 1.3 Hz, H^c^), 7.32 (2 H, td, *J* 7.5, 1.2 Hz, H^b^ and NH), 4.71 (2 H, d, *J* 6.4 Hz, CH_2_), 1.35 (12 H, s, 4 × CH_3_). Minor conformer: Only selected signals visible. 7.81 (1 H, d, *J* 6.7), 7.77 (1 H, d, *J* 7.4), 4.70 (2 H, s, CH_2_), 1.35 (12 H, s, 4 × CH_3_). δc (CDCl_3_, 100 MHz): 163.5 (C=O), 143.6 (CH_2_*C*q), 136.9 (C^a^), 136.1, 134.6, 133.6, 132.9 (4 × Cq), 132.0 (C^c^), 130.3 (C^d^), 128.7 (C^e^), 127.4 (C^b^), 125.7 (Cq), 84.4 (2 × pinacol Cq), 44.8 (CH_2_) and 25.1 (4 × pinacol CH_3_); note, Ar*C_q_*-B is not visible. δ_B_ (dissolved in CDCl_3_, 96 MHz): 31.8 ppm. X-ray Crystallographic analysis.


**4,5,6,7-Tetrachloro-2-(4-(4,4,5,5-tetramethyl-1,3,2-dioxaborolan-2-yl)benzyl)isoindoline-1,3-dione para 8**




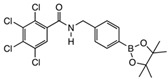



Tetrachlorophthalimide **6** (352 mg, 1.235 mmol, 1.0 equiv.) and 4-(bromomethyl)phenylboronic acid pinacol ester **para 7** (440 mg, 1.474 mmol, 1.2 equiv.) were stirred in *N*,*N*-dimethylformamide (8 mL). Sodium hydride (123 mg, 12.80 mmol, 2.5 equiv.) was then carefully added as hydrogen gas was given off to give a pale creamy yellow reaction mixture and heated to 100 °C. After 48 h the reaction mixture was cooled to room temperature and a golden honey colour solution with a golden yellow precipitate was visible. Two drops of R.O. water were added to quench and evaporated. The residue was dissolved in chloroform (30 mL) and gravity filtered. The filtrate was evaporated to give **para 8** (339 mg, 56%). M.p. 110–112 °C. *m*/*z* (HRMS ES^+^): Relative intensities for C_21_H_19_BCl_4_NO_4_, [M + H]^+^, 473.04016 (20%), 474.03632 (74%), 475.03815 (39%), 476.03317 (100%), 477.03586 (31%), 478.02997 (47%), 479.03320 (13%), 480.02664 (10%); calculated 473.03994 (18%), 474.03667 (78%), 475.03816 (41%), 476.03372 (100%), 477.03587 (33%), 478.03042 (46%), 479.03320 (13%), 480.02747 (10%). ν_max_ (thin film, cm^−1^): 3265 (m, br, NH), 2976, 2926, 2853 (m, alkyl CHs), 1652 (s, C=ONH, amide I), 1575 (w, ArC=C), 1517 (m, C=ONH, amide II), 1459 (m, C-N), 1407 (m, C-B), 1358, 1340, 1088 (s, sp^2^ B-O), 751 (m, C-Cl), 657 (s, C-B(-O)_2_, out of plane bending). δ_H_ (CDCl_3_, 400 MHz): 7.81 (2 H, d, *J* 8.0, 2 × ArH), 7.65 (1 H, s, H^e^), 7.36 (2 H, d, *J* 8.0 Hz, 2 × ArH), 6.32–6.40 (1 H, br s, NH), 4.65 (2 H, d, *J* 5.6 Hz), 1.35 (12 H, s, 4 × CH_3_). δc (CDCl_3_, 400 MHz): 135.5 (2 × Ar-CH), 128.6 (C^e^), 127.4 (2 × Ar-CH), 84.1 (2 × pinacol Cq) and 25.0 (4 × pinacol CH_3_); note, Cq are not visible. δ_B_ (dissolved in CDCl_3_, 96 MHz): 31.6 ppm.

## 7. Conclusions

We reported an expedited synthesis to a small library of novel borylated 2,3,4,5-tetrachlorophthalimides and 2,3,4,5-tetrachlorobenzamides. Biological assays against glycosidase enzymes and cancer cell lines highlighted a good inhibitor for bovine liver β-galactosidase and three potent growth inhibitors and, of these, one selective growth inhibitor for cancer versus healthy cell lines in the cancer assay. These drugs are set for further derivatisations and utilisation in BNCT.

## Data Availability

Not applicable.

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
