# Peer review of "Borylated 2,3,4,5-Tetrachlorophthalimide and Their 2,3,4,5-Tetrachlorobenzamide Analogues: Synthesis, Their Glycosidase Inhibition and Anticancer Properties in View to Boron Neutron Capture Therapy"

_molecules, 2022, doi:10.3390/molecules27113447_

Round 1

Reviewer 1 Report

Simone and co-workers describe in this paper the synthesis of 2,3,4,5-tetrachlorophthalimides and 2,3,4,5-tetrachlorobenzamides bearing a methylphenylboronic acid pinacol ester, or a phenylboronic acid pinacol ester group bonded to the nitrogen atom. These compounds were prepared following two different strategies. Double acylation of borylated benzylamines or anilines with tetrachlorophthalic anhydride led to tetrachlorophthalimides and tetrachlorobenzamides, respectively. Meanwhile, reaction of tetrachlorophthalimide with borylated benzyl bromide, after deprotonation with sodium hydride, produced N-benzylborylated 2,3,4,5-tetrachlorobenzamides. In all cases, decarboxylation or decarbonylation occurred when benzamides are the reaction products. In addition to the synthesis, the main body or this study is based on the biological evaluation of these compounds as glycosidase inhibitors, and antiproliferative activity in tumor cell lines. It was found that the three synthesized benzyl phthalimides did not display glycosidase inhibition. However, benzamides showed from good to moderate selectivity in the inhibition of a panel of 15 glycosidases. Some of these glycosidases are involved in diabetes and viral infections. The in vitro activity of these compounds against 10 human cancer lines and MCF10A line (normal breast line) was also texted. It was found that tetrachlorophthalimide para 3 and tetrachlorobenzamide ortho 5 were extremely potent antiproliferative agents against most of the cancer cell lines, and tetrachlorophthalimide meta 3 showed lower percent cell growth inhibitions. However, tetrachlorobenzamide ortho 5 did not affect much the growth of normal cell line MCF10A. Due to this selectivity, ortho 5 could be considered a promising candidate for further studies in boron neutron capture therapy (BNCT).

Experimental procedures (very simple reaction conditions) and characterization by physical (melting points) and spectroscopic means (HRMS, IR, 1H- and 13C-NMR) of seven compounds [tetrachlorophthalimides (2) and tetrachlorobenzamides (5)] are reported in the Experimental section. In summary, I found this study of some interest for Medicinal Chemists and, therefore, I recommend the publication of this paper in Molecules after addressing the following comments and some minor remarks that should be corrected or clarified.

  1. Page 1, last sentence of the Abstract section: “…a greener synthetic access to such structures is described”. What does “greener” mean here? Apparently, were this type of compounds previously synthesize under harsh conditions using toxic reagents and solvents? By the way, all reactions are carried out in acetonitrile, which is not a green solvent.  
  2. Authors use the term “small library” referring to 5 (benzamides) or 7 (benzamides + phthalamides) compounds. I think it is not correct to refer to a small library as a group of 5 or 7 compounds.
  3. The authors complied to be much more concise in “2.2. Decarbonylation reaction”, “2.3. Decarboxylation reaction to benzamides” and “3. X-Ray Crystallography Commentary” sections. These sections are too long, and of no interest to potential readers. In addition, X-ray crystallographic data are provided as Supporting Information.
  4. Authors should also provide copies of 1H- and 13C-NMR spectra for compounds 3, 5 and 8 in the Supporting Information.
  5. Typos:

5.1. Recurring error throught the text, “nhibitors” must be “inhibitors”

5.2. Authors should pay attention to spaces in heading sections. For instance, on page 8, “4.2.2.2,3,4,5-. Tetrachlorophthalimides” must be “4.3.2. 2,3,4,5-Tetrachlorophthalimides”. On page 9, “4.3.2.2,3,4,5-. Tetrachlorophthalimides” must be “4.3.2. 2,3,4,5-Tetrachlorophthalimides”

5.3. Page 9, 4.3.2. section: “para 3” and “meta 3” should be in bold for consistency throughout the text.

5.4. Page 11, line 15: “… non toxic toxic in the…”

5.5. Page 12, line 1: “…and their their…”

Reviewer 2 Report

Very interesting results by Michela I. Simone.

This borylated 2,3,4,5-tetrachlorophthalimide and their 2,3,4,5-tetrachlorobenzamide analogues synthesis are remarkable and allow to be achieved by using less chromatography work.

The study is detailed, well designed, and executed. Also, the report is clear and the compounds pure and well characterized.

In conclusion, a good manuscript, with novel chemistry, non-trivial, with high levels of novelty and useful for the synthetic community.

I would recommend publication, essentially as it is, although

I have one question for authors, on the page 2 and 3, From Synthetic Strategy 1:

Why was gravity-filtered to remove the precipitate? Why not use the vacuum filtration?
